# Medium and High Phosphorous Ni-P Coatings Obtained via an Electroless Approach: Optimization of Solution Formulation and Characterization of Coatings

**Virgilio Genova** [1,*], **Laura Paglia** [1], **Giovanni Pulci** [1], **Giulia Pedrizzetti** [1], **Alice Pranzetti** [2], **Marco Romanelli** [2] and **Francesco Marra** [1]

1    Department of Chemical Engineering, Materials, Environment, Sapienza University of Rome, INSTM Reference Laboratory for Engineering of Surface Treatments, Via Eudossiana 18, 00184 Rome, Italy; laura.paglia@uniroma1.it (L.P.); giovanni.pulci@uniroma1.it (G.P.); giulia.pedrizzetti@uniroma1.it (G.P.); francesco.marra@uniroma1.it (F.M.)
2    Nuovo Pignone International, Baker Hughes, Via Felice Matteucci 2, 50127 Florence, Italy; alice.pranzetti@bakerhughes.com (A.P.); marco.romanelli@bakerhughes.com (M.R.)
*    Correspondence: virgilio.genova@uniroma1.it; Tel.: +39-(0)644-585-314

**Abstract:** A new lead-free electroless Ni-P plating solution was developed for the deposition of coatings with medium phosphorus content (MP, 6–9 wt%), and its composition was optimized to obtain deposits with high phosphorus (HP, 10–14 wt%). Cleaning and activation treatments were studied in terms of effectiveness and influence on the deposition rate. The concentration of reagents (nickel salt, complexing agent, reducing agent and stabilizer) was studied, and their combined effect on P content and plating rate was investigated. The obtained coatings were analyzed by SEM and XRD and thermally treated at 400 °C and 600 °C to study microstructural evolution. Vickers hardness was measured on as-deposited and annealed coatings to relate hardness evolution to microstructural changes after thermal treatments. Optimal deposition conditions were determined, enabling the production of MP coatings (6.5 wt% P) with a plating rate of 40 μm/h and HP coatings (10.9 wt% P) with a plating rate of 25 μm/h at 90 °C. Samples heat-treated at 400 °C showed improved hardness thanks to crystallization and microprecipitation of $Ni_3P$ hard phases, whereas hardness decrease was observed after treatment at 600 °C due to the combined effect of grain growth and coarsening of $Ni_3P$ precipitates. No through-the-thickness cracks were detected by the Ferroxyl reagent after heat treatments.

**Keywords:** electroless Ni-P; electroless deposition; microhardness; annealing; plating parameters

## 1. Introduction

Electroless deposition is an easy and cost-effective method for metal plating that consists of the reduction of metal ions in aqueous systems without the use of external electric current [1]. The electrons involved in the reduction reaction come from the oxidation of a reducing agent in solution. The redox process and the consequent coating growth take place on a catalytic substrate immersed in the solution. This plating technique is often referred to as autocatalytic since the growing metallic coating keeps on catalyzing the reaction and allows the continuing deposition of thick layers. Compared with the electrodeposition method, electroless plating guarantees several advantages that make it a promising technique for applications in the energy production, aerospace and automotive industries [2–4]:

- It is a non-line-of-sight technique, and the coating forms on any part of the catalytic surface placed in contact with the plating solution.
- No electric field participates in the reduction process, and a uniform, conformal and homogeneous coating can be obtained on any geometry, regardless of its shape.

- The flexibility of the solution chemistry allows its tuning to investigate innovative and cheaper formulations, to meet the specific needs of the final product [5,6].

The Ni-P system, often reported as ENP (electroless nickel plating) for simplicity, represents the most widespread technology of electroless plating: the use of hypophosphite ion as the reducing agent leads to the co-deposition of phosphorus in the nickel matrix, giving a coating with excellent properties of hardness, solderability and corrosion resistance. For this reason, Ni-P coatings are widely used to protect engineering components from surface degradation when operating environments are rich in contaminants and deteriorating agents [2,7–10]. Ni-P coatings can be classified according to the amount of phosphorous in the matrix: low phosphorous (LP, 3–5 wt% of P), medium phosphorous (MP, 6–9 wt% of P) and high phosphorous (HP, 10–14 wt% of P) [11–13]. The P quantity strongly influences coating properties by modifying the microstructure of the metallic alloy: LP coatings are crystalline, MP typically shows a mixed amorphous-crystalline microstructure, and HP is amorphous [14].

The mechanism of electroless Ni-P deposition has been extensively studied [15,16]; however, mechanisms involved in the electroless deposition process are still being discussed, and many aspects of the process remain unexplored. In any of the proposed studies, the principles governing Ni deposition are established by analyzing the main constituents of the plating solution, which are:

- Nickel source: A soluble, hydrated and stable $Ni^{2+}$ salt. The use of sulfate and acetate gives the same results in terms of coating quality, whereas using chloride negatively influences the corrosion resistance of the produced coatings [17]. Sulphate is the most used since it is generally cheaper than acetate.
- Reducing agent: Sodium hypophosphite represents the most studied and used reducing agent [1,18].
- Complexing agent/buffer solution: The introduction of organic acids or their salts have a double function: first, they guarantee a buffer action that stabilizes the pH of the solution; second, the organic base can complex the nickel ions in solution and limit their reactivity, thus increasing bath stability. The ENP process is called autocatalytic because the substrate itself catalyzes the oxidation of the reducing agent and the reduction of the metal cations present in the solution [19]. The deposition reaction usually occurs along with the formation of byproducts that make the deposition process uncontrolled, and chemical stabilization is necessary to avoid the decomposition of the ENP plating bath. In most commercial ENP solutions, the extended service life of the bath is guaranteed by lead addition in ppm [20]. Despite this strategy being remarkably effective, the presence of $Pb^{2+}$ in the solution causes health hazards and high costs the waste-solution disposal. The study of environmentally friendly alternatives to lead stabilizers is nowadays an important goal to improve ENP applicability and meet the strict environmental regulations towards a more sustainable development. This work presents the formulation, the study and the characterization of lead-free solutions capable of depositing medium phosphorus coatings and their optimization to increase the amount of co-deposited phosphorus to obtain HP coatings. Changes in the chemistry of the plating solution make it possible to manufacture Ni-P coatings with different P content in the alloy, according to the properties required for the specific applications. As an example, Table 1 reports different $Ni^{2+}/H_2PO_2^-$ molar ratios and the consequent wt% of P in the matrix obtained from different recipes presented in a list of works from 1994 to 2020. It can be noted that a similar molar ratio of nickel sulfate and sodium hypophosphite does not have an obvious relation to the content of P inside the matrix. This is because the P quantity does not depend only on the $Ni^{2+}/H_2PO_2^-$ ratio but on the whole system employed for the deposition, including temperature, stabilizers and complexing agents. Although all these parameters have been extensively studied, to the authors' knowledge, there are currently no works in literature that systematically relate the concentrations of the various reagents to the properties of the coatings. The tailoring of the P content by the change in concentra-

tion of reagents in the deposition solution can be a desirable aspect to increase the flexibility and applicability of specific electroless Ni-P baths. Moreover, a comprehensive understanding of the cross-relations between reagents is necessary for a better comprehension of their effect on deposition and coating properties.

**Table 1.** Examples of different $Ni^{2+}/H_2PO_2^-$ molar ratios used to produce a coating with different P content. --- means that the wt% *p*-values were not indicated.

| | Abrantes et al. (1994) [15] | Keping and Fang (1997) [21] | Lin and Hwang (2002) [22] | Chen et al. (2002) [23] | Cheong et al. (2004) [23] | Baskaran et al. (2005) [24] | Liu et al. (2008) [25] | Rahimi et al. (2009) [26] | Wu et al. (2019) [27] | Park and Kim (2019) [28] | Lin and Chou (2020) [29] |
|---|---|---|---|---|---|---|---|---|---|---|---|
| $Ni^{2+}/H_2PO_2^-$ | 0.88 | 0.82 | 2.06 | 0.36 | 0.60 | 0.35 | 0.39 | 0.44 | 0.35 | 0.56 | 0.89 |
| wt% P | 12 | --- | 8 | 10 | 10 | 11 | 10.5 | 9 | 12 | 3–6 | 5–8 |

For this reason, the first part of this work is focused on the investigation and correlation of different concentrations of reagents in order to methodically study their influence on P content and plating rate. Optimization of bath chemistry and deposition parameters was performed to define a recipe that can be applied for the deposition of both MP and HP coatings. Isothermal heat treatments at 400 °C and 600 °C were performed in the air in the produced samples to assess properties modification induced by the microstructure evolution. The stability and durability of the plating solution are guaranteed using an organosulfur compound as a stabilizer, and the obtained coatings were characterized in terms of composition, morphology, hardness and adhesion in the as-deposited condition and after annealing at 400 °C and 600 °C in air. The long-term goal is to provide a safe and environmentally friendly ENP solution for the deposition of MP and HP coatings that can eventually be considered a valid alternative to the more widespread lead-based plating baths.

## 2. Materials and Methods

### 2.1. Surface Preparation

All the depositions were carried out on 15 × 15 × 3 mm samples of ASTM 182 F22 steel. The procedures for surface pre-treatment and cleaning follow the B 183-79, B 322-99 and B 733-97 ASTM international standards. These procedures are necessary for removing tarnish, light rust, grease, contaminants and oxides before immersion of the specimen in the electroless plating solution. The substrates to be coated can be cleaned from grease with an alkaline solution, while the activation of the metal (i.e., removal of surface oxides that hamper deposition) is usually carried out in acidic baths. Adequate cleaning is fundamental to guarantee activation of the deposition process, and the steps of the substrate preparation procedure were defined and optimized to obtain adequate deposition conditions:

- Pre-cleaning: Performed to remove oils, lubricants or massive oxidation products that can hinder the deposition by reducing the exposed autocatalytic surface. The effect of soak cleaning and/or sandblasting was investigated. In particular, soak cleaning was carried out in a 1 M NaOH solution at 80 °C for 10 min.
- Sandblasting was used to remove the residuals from the soaked cleaning and to increase the roughness of the surface; it was performed by using corundum (mesh 80) as abrasive material.
- Acid pickling: The effect of pickling with HCl 37% at different concentrations (30, 40, 50 and 100 vol.% of HCl 37 wt%) was studied to evaluate the best activation solution. The tests were carried out on samples in the as-received condition (AR). Heat treatments in the air at 400 °C and 600 °C for 4 h were carried out to obtain a thermally grown oxide layer and simulate a surface that is not intrinsically active to electroless deposition and evaluate the effectiveness of the different activation solutions. Acid pickling was performed by immersing specimens in 100 mL of acid solution at room temperature for 1 min.

- Water rinse: After every step of the activation procedure, samples were rinsed with deionized water (DW) in an ultrasonic bath for 1 min to remove every contaminant or residue.

The success of the activation procedure was evaluated by submerging specifically pre-treated samples in the plating solution and observing the appearance of $H_2$ evolution (effervescence) that witnesses the proceeding of the electroless reaction. The effectiveness of activation was evaluated by assessing the deposition rate after different surface pre-treatment procedures, with higher rates determining the best results.

### 2.2. Solution and Coating Preparation

All chemicals were purchased from Alfa Aesar (Thermofisher Scientific, Kandel, Germany) and used without any further purification. Two recipes were optimized to obtain Ni-P coatings with different P concentrations (namely, MP and HP). All the solutions were prepared with the same procedure: the reagents were added separately and in a specific order in deionized water, and each reagent was added only after the complete dissolution of the previous one. Sodium hypophosphite was added as the first reagent; then, sodium acetate was added to stabilize the pH. Citric acid was added to set up the coordination of the $Ni^{2+}$ that is added last as nickel sulfate. The thio-organic compound was added in the final step of the preparation procedure. Once the solution was complete, the reactor was covered and heated on a hot plate (IKA™ (Stauten, Germany) RET Control-Visc. equipped with an external temperature sensor PT1000 for continuous monitoring of the bath temperature.

The plate uses a PID (proportional–integral–derivative), controller able to maintain the target temperature with a precision of about 0.5 °C. This value is important to keep the solution stable during the deposition time. The solution is kept in constant mechanical agitation using an overhead stirrer Heidolph Hei-TORQUE Core (Heidolph Instruments GmbH & Co. KG, Schwabach, Germany) equipped with PTFE coated impeller. pH was constantly monitored using a METTLER TOLEDO™ Seven Excellence pH-meter model S400, equipped with pH sensor InLab® Viscous Pro-ISM (Mettler Toledo, Columbus, OH, USA). Temperature, agitation and pH value were continuously controlled during the deposition. A recipe for the production of MP coating was first optimized (with formulation reported in Table 2); after that, the quantities of the nickel source ($NiSO_4$) and the complexing agent (citric acid) were modified in order to obtain a formulation for HP coatings. The properties of the coatings obtained with the modified formulation were analyzed considering both the variation in the concentration of a single reagent and the combined effect of the concentration change of both the nickel source and the complexing agent. The buffer quantity (sodium acetate) has not been changed in order to maintain the pH constant at a value of 4.2. It is well known that pH strongly influences the P content and the plating rate since it induces a shift in the equilibrium of reactions, and its effect was extensively studied in the literature [1]; variation of the pH values implies the use of a different buffer agent that would invariably influence the solution formulation. For this reason, the buffer reagent and its concentration were excluded from the studied parameters.

**Table 2.** Medium phosphorous (MP) bath formulation.

| Function | Name | Chemical Formula | MP (g/L) |
|---|---|---|---|
| Reducing agent | Sodium hypophosphite | $NaH_2PO_2$ | 70 |
| Buffer | Sodium acetate | $C_2H_3NaO_2$ | 15 |
| Chelating Agent | Citric acid | $C_6H_8O_7$ | 7 |
| Source of Nickel | Nickel sulfate | $NiSO_4$ | 12 |
| Stabilizer * | Thio-organic compound (TOC) | R-CS | 5 (ppm) |

* The stabilizer was added by liquid solution (1 mol/kg) and respecting the quantity in ppm.

The results regarding the quality of the coating are presented in terms of plating rate (thickness measured by cross-section SEM analysis) and composition. The P content in the coating was measured by EDS on cross-section micrographs. Analysis was performed on at least five areas comprising 70% of the coating thickness, starting from the top of the coating. Despite the narrow sensitivity of the instrument, EDS was selected as a technique to determine the P quantity as reported in several works, because it is widely recognized as a technique for determining the amount of phosphorus in an electroless Ni-P system [15,22,24,27,28,30,31]. All the coated samples, unless otherwise stated, were thermally treated (right after the deposition) at 180 °C for 2 h in air to avoid hydrogen embrittlement phenomena [17,32,33]: such coatings will be referred to as as-coated hereafter. Isothermal heat treatments at 400 °C and 600 °C for 1 h were carried out on all the produced samples in order to study the evolution of properties upon crystallization. Variation of crystallite size with thermal treatments was calculated using Scherrer's equation [34]:

$$D = \frac{0.94\lambda}{\beta \cos(\theta)} \, , \tag{1}$$

where $\lambda$ is the wavelength of the radiation used, $\beta$ is the half-maximum width, and $\theta$ is the position of the main peak. No correction for instrumental broadening was made.

All treatments were performed in the air in a Lenton tube furnace (now Carbolite Gero Ltd., Sheffield, UK) with a heating rate of 10° C/min.

### 2.3. Coating Characterization

Surface morphology, thickness and composition of the coatings were determined with a FEG-SEM Tescan Mira3 (Tescan, Brno, Czech Republic) equipped with Edax Octane Elect EDS system detector (Edax/Ametek Inc. Pleasanton, CA, USA) for energy-dispersive X-Ray spectroscopy (EDS); Edax Team v.4.5 software was employed for the elementary analysis. The top view observation is important to evaluate changes in the surface morphology and microstructure according to the different deposition parameters, while the observation of cross-sections allows measurement of the coating thickness, evaluation of the deposition rate and assessment of the P content (expressed in wt%) through coating thickness. Samples were prepared for cross-sectional analysis by mounting in epoxy resin, with subsequent cut with a slow-speed precision saw and polished with SiC papers (grit P400 to P1200) and diamond suspensions (up to 1 μm). X-ray diffraction analysis (XRD) was performed with a Philips X'Pert device (PANalytical B.V., Almelo, The Netherlands) to identify the crystalline phases in the coatings. The XRD device operated at 40 kV and 40 mA with CuKα1 radiation, a scan range of 30–80° (2θ), a step size of 0.02° and a counting time of 2 s.

Mechanical characterization tests were performed on the optimized MP and HP coatings in the as-coated condition and after heat treatments at 400 °C and 600 °C. Vickers microhardness was measured according to ASTM E384-11 using a LEICA VMHT (Leica GmbH, Wetzlar, Germany) equipped with a Vickers diamond indenter (load 50 gf, time 15 s). The distance between two indentations was ≥50 μm, and results report the average and standard deviation of at least fifteen measurements for each coating. Adhesion strength of coatings was measured with an Instron 5584 (Illinois Tool Works Inc., Norwood, MA, USA) mechanical testing machine equipped with a load cell of 150 kN at a crosshead rate of 2 mm/min according to the ASTM C633. Each specimen was obtained by assembling one cylindrical coated substrate to the sandblasted faces of the loading fixtures by an adhesive bonding agent (Polyamide-epoxy FM 1000 Adhesive Film, CyTech, Treviso, Italy); a self-aligning device was adopted to guarantee a perfect co-axiality of the two parts during the assembly and the heat treatment necessary for the cure of the glue (180 °C for 1 h).

The presence of through-the-thickness porosities or cracks was detected by the ferroxyl reagent test according to the procedure of ASTM B689-87: a filter paper was placed on the sample, and some drops of the ferroxyl solution were poured on it with a Pasteur pipette. The ferroxyl solution consists of 10 g/L potassium hexacyanoferrate (III), 60 g/L sodium chloride, 20 mL of phenolphthalein for 1 L of deionized water. The solution turns blue

when $Fe^{2+}$ ions are detected: if blue spots appear on the filter paper, it means that the solution reached the surface of the steel substrate, and the test can be considered positive, revealing the presence of through-the-thickness porosities or cracks.

## 3. Results

### 3.1. Surface Preparation

The formation of a thin oxide layer on the surface of the specimens can hind the activation of the deposition reaction since the substrate is required to be conductive to trigger the oxidation reaction. For this reason, specific and effective surface pre-treatments are important steps to avoid inhomogeneities in the first layer of the coating. The effect of surface pre-deposition treatments (referred to as pre-treatments) was studied by varying the conditions of acid pickling (concentration of HCl 37% was varied from 30 vol.% to 100 vol.%), and results are reported in Table 3. Pre-treatments were performed on received (AR) substrates and on heat-treated substrates (at 400 and 600 °C for 1 h in air). Samples are reported as "Activated" whether the deposition was successful, and, in such cases, the deposition rates are reported in the last column of Table 3. When tests are performed on AR substrates, two cases do not guarantee activation: when 10 min lab air exposure follows the pre-cleaning process (soak cleaning in NaOH and sandblasting) and when acid pickling is performed with 100 vol.% of HCl 37 wt%. In the first case, the pre-cleaning effectively degreases the substrate, and sandblasting removes initial oxides, but ferrous substrates exhibit low corrosion resistance, and 10 min of lab air exposure is sufficient to cause further oxidation and hinder deposition. In the case of HCl 100 vol.%, the acid treatment for pickling can be considered too aggressive and leads to the formation of corrosion products that inhibit the deposition process [35]. This last phenomenon, known as over-corrosion, can sometimes be observed when substrates are cleaned by pickling with aggressive acids [36]. The reactions involved in the acid pickling process can be summarized in the following reactions:

$$Fe_3O_4(s) + 8H^+(l) \rightarrow 2Fe^{3+}(l) + Fe^{2+} + 4H_2O(l) \tag{2}$$

$$Fe_2O_3(s) + 6H^+(l) \rightarrow 2Fe^{3+}(l) + 3H_2O(l) \tag{3}$$

$$FeO(s) + 2H^+(l) \rightarrow Fe^{2+}(l) + H_2O(l) \tag{4}$$

$$Fe(s) + 2H^+(l) \rightarrow Fe^{2+}(l) + H_2(g) \tag{5}$$

Equations (2)–(4) refer to the removal of oxides from the metal surface, while Equation (5) is related to the phenomenon of over-corrosion, which causes the dissolution of Fe and the release of Fe ions in solution. If the pickling process is performed using an excessively strong acid, like HCl 100 vol.%, the removal of the oxides that naturally grow on the surface is so fast that some of the pickling solution reacts with the base metal, making it more prone to oxidation when exposed to air. The XRD analysis before and after the activation treatments was performed to better investigate the formation of products that prevent effective activation of the substrate. Figure 1 shows a comparison of XRD spectra of the as-received substrate (i) before the removal of oil and lubricants that prevent oxidation during the storage, (ii) after the pre-cleaning process (soaking in NaOH 1 M at 80 °C for 1 h and sandblasting) and exposure to 10 min lab air, (iii) after treatment with HCl 100 vol.% and exposure to 10 min lab air and (iv) after treatment with HCl 40 vol.% and exposure to 10 min lab air. All samples were protected with Cortec® corrosion inhibitor oil before XRD analysis to prevent further air exposure. Peaks at 2θ position 45° and 65° are referred to as Fe (JCPDS no. 87-0722). As expected, only Fe peaks are present in the AR samples. Considering the pre-treated substrates, both the AR samples after pre-cleaning and after acid pickling with HCl 100% show $Fe_3O_4$ peak (magnetite, JCPDS no. 76-0955), which confirms the presence of an inert oxide layer that prevents activation of the deposition. Conversely, no magnetite peaks are detected on the XRD spectrum of the sample

treated with HCl 40 vol.%. This suggests that both sandblasting and over-corrosion make the substrate more vulnerable to oxidation, probably due to the larger exposed surface, whereas samples that underwent pickling with HCl 40 vol.% require a longer amount of time to oxidate and show the best results in terms of activation when immersed in the plating solution.

**Table 3.** Activation results after acid pickling. AR refers to the samples in the as-received condition; HT400C refers to the samples oxidized at 400 °C for 4 h; HT600C refers to the samples oxidized at 600 °C for 4 h.

| Sample | HCl Concentration (%vol) | Result | Plating Rate (mg/cm$^2$/h) |
|---|---|---|---|
| AR | (Pre-cleaning and 1 h air exposure, no activation procedure) | Not Activated | - |
| | 30 | Activated | 8.60 |
| | 40 | Activated | 15.7 |
| | 50 | Activated | 14.0 |
| | 100 | Not activated | - |
| HT 400 °C | (Pre-cleaning and 1 h air exposure, no activation procedure) | Not Activated | - |
| | 30 | Activated | 13.8 |
| | 40 | Activated | 12.9 |
| | 50 | Activated | 15.1 |
| | 100 | Activated | 11.0 |
| HT 600 °C | (Pre-cleaning and 1 h air exposure, no activation procedure) | Not Activated | - |
| | 30 | Not activated | - |
| | 40 | Not activated | - |
| | 50 | Not Activated | - |
| | 100 | Activated | 9.80 |

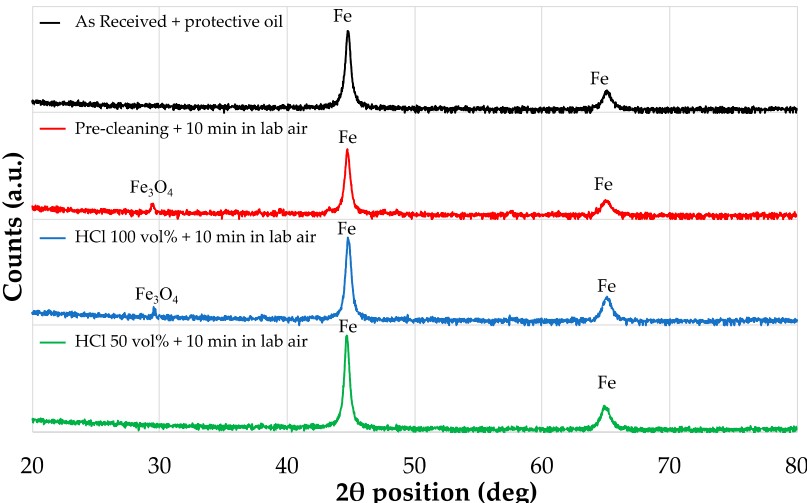

**Figure 1.** Comparison of XRD spectra of as received sample (black), after pre-cleaning and 10 min lab air exposure (red), after acid picking with HCl 100 vol.% and 10 min lab air exposure (blue) and after acid picking with HCl 50 vol.% and 10 min lab air exposure (green).

Therefore, the success of the activation procedure strongly depends on the balance between the strength of the acid for pickling and the amount of oxide on the surface of

the substrate (the degradation of the substrate and the amount of the oxide on the surface strongly depends on the preservation and storage conditions of the components). To better understand this phenomenon, the substrates were oxidized by exposure at 400 °C and 600 °C for 4 h in the air using a Lenton tube furnace (now Carbolite Gero Ltd., Sheffield, UK) to simulate different conditions of degradation. The specimens treated at 400 °C are expected to exhibit a more pronounced surface oxidation than the AR, and results in Table 3 show that successful activation can be obtained for all concentrations of HCl. Conversely, the substrates treated at 600 °C are oxidized to the extent that only pickling with undiluted HCl can provide activation. The strength of the acidic solutions used for the activation treatment also influences the plating rate: the slower deposition associated with lower concentrations of HCl can be addressed to incomplete pickling, which is only capable of removing some part of the oxide scale. This leads to smaller exposure of the catalytic surface and a lower number of nucleation sites catalytically active for deposition. In such cases, the Ni-P coating mainly forms by lateral growth of the present nucleation germs and the plating rate is globally slower. The best results are therefore obtained when achieving a good balance between the HCl concentration of the acidic solution and the amount of oxide to be removed.

The XRD analysis of the activation tests on the substrates heat-treated at 400 °C and 600 °C are reported in Figure 2a and 2b, respectively. The spectra of the samples just after heat treatment reveal that the high temperate exposure leads to the formation of $Fe_2O_3$ (hematite, JCPDS no. 73-2234). Either the pre-cleaning process (here given by sandblasting only) and the acid pickling with HCl 100% are sufficient to remove the oxide grown upon thermal treatments, so hematite is no more detectable from XRD; nevertheless, just like in the case of AR samples, sandblasting and acid pickling with HCl 100 vol.% expose the base metal, causing the formation of a layer of magnetite layer when exposed to air. However, the activation tests demonstrate that the amount of over-corrosion is not sufficient to totally prevent activation and deposition proceeds at a slower rate.

Figure 3 shows the schematic representation of the acidic pickling process with the two extremes where activation is not obtained: the "soft acid" refers to the case in which the pickling does not remove the oxide scale from the substrate so that the catalytic surface is not exposed and the deposition is hampered; the "hard acid" case occurs when the base metal is over-corroded, and the surface is more vulnerable to oxidation, causing formation of products that can hinder (or slow down) deposition.

According to the experimental activity, the optimal combination of soak cleaning, sandblasting and acid pickling was selected as the standard activation procedure and reported in Table 4.

**Table 4.** Standardized procedure of surface pre-deposition activation treatment.

| Treatment | Time |
|---|---|
| NaOH-1 M-80 °C | 10 min |
| Water rinsing | - |
| Sandblasting | - |
| Sonication in $H_2O$ | 5 min |
| HCl 50 %vol | 1 min |
| Water rinsing | - |
| Plating | - |

The sonication step in deionized water (DW) was added since it is necessary to remove the abrasive material that remains stuck in the substrate after sandblasting.

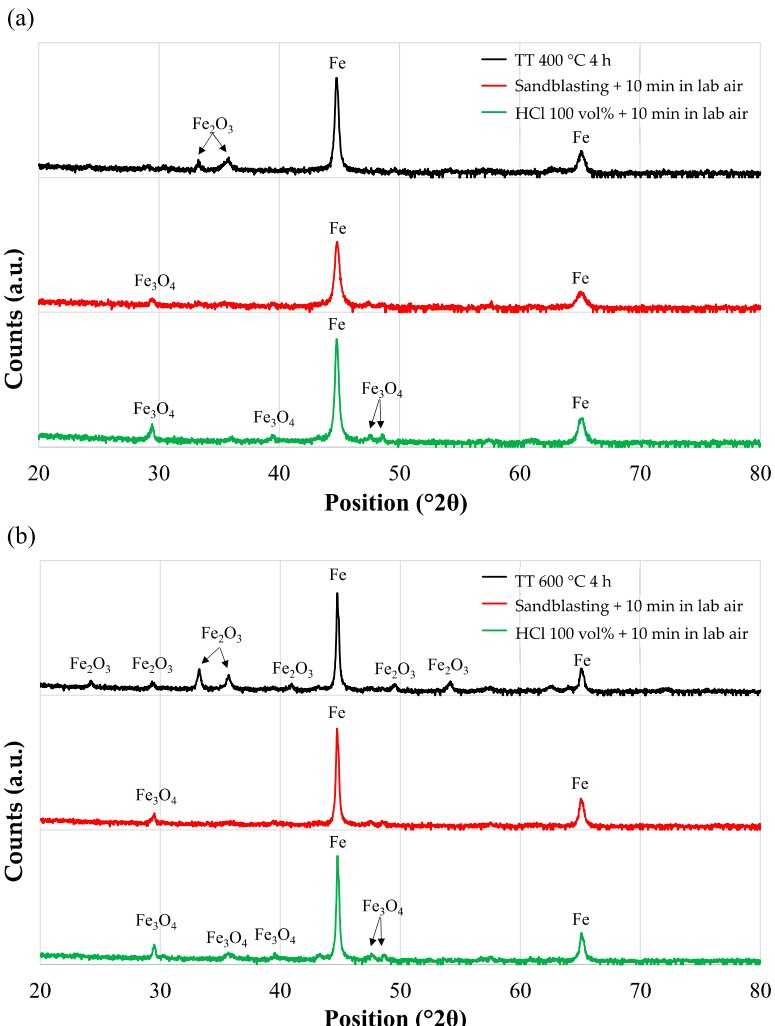

**Figure 2.** (**a**) XRD spectra of samples oxidized at 400 °C for 4 h: without any treatment (top), after sandblasting + air lab exposure of 1 h (center) and after acid pickling with HCl 100 vol.% (bottom) (**b**) XRD spectra of samples oxidized at 600 °C for 4 h: without any treatment (top), after sandblasting + air lab exposure of 1 h (center) and after acid pickling with HCl 100 vol.% (bottom).

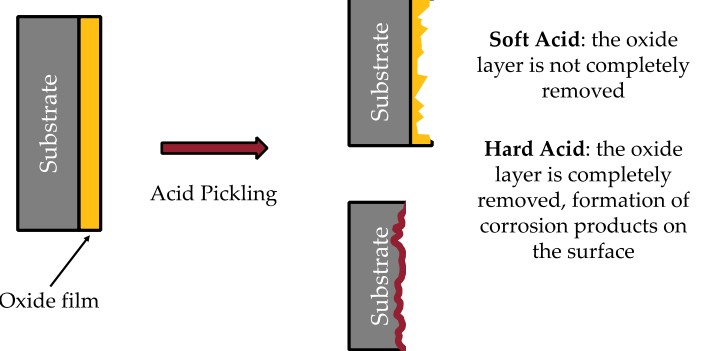

**Figure 3.** Acid pickling effects the oxide layer in the case where activation is not achieved.

### 3.2. Study of the Solution

The influence of temperature on the deposition rate was investigated for the MP plating solution (Figure 4). Only temperatures that guarantee a deposition rate higher than 5 µm/h are reported in the graph, but this limit does not represent the lower temperate of reaction activation. Acidic baths with comparable formulations are reported in the

literature to be active at temperatures as low as 50 °C [37], but too-slow deposition rates may limit applicability and increase costs when thick functional coatings are required (e.g., anti-wear applications). Temperatures higher than 95 °C have also been excluded to avoid significant evaporation of the plating solution and the consequent concentration of the reagents. EDS analysis confirmed that the plating temperature does not influence the %P for MP deposition, as similarly reported in other studies [35]. Considering these results, the selected operating temperature was 90 °C.

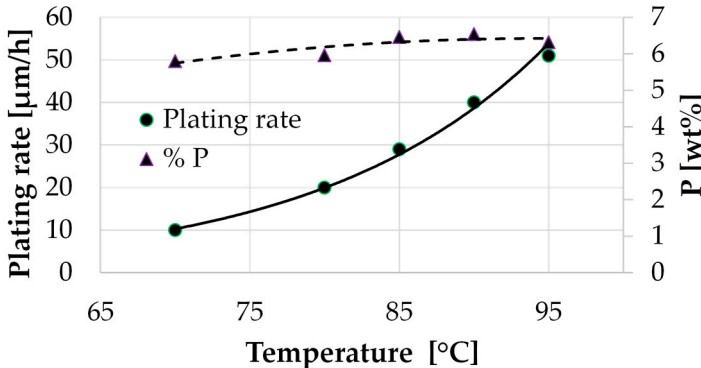

**Figure 4.** Plating rate and %P as a temperature function for MP plating solution. Dashed and solid lines are inserted for the readability of the graph and do not represent any mathematical relation.

Given the stability and high plating rate of MP formulation, the concentration of reagents was optimized in order to increase the amount of co-deposited phosphorus, with the final aim of producing HP coatings.

The first studied reagent was $NiSO_4$ since it represents the source of nickel-free ions in solution. Reducing its quantity, without any variation on the concentration of the complexing agent and the reducing agent, theoretically entails the decrease of deposition rate due to the lower $Ni^{2+}$ availability [38], according to the reaction reported in Equation (6) [31]:

$$Ni^{2+} + 2H_2PO_2^- + 2H_2O \rightarrow Ni^0 + 2H_2PO_3^- + 2H^+ + H_2 \tag{6}$$

This trend is confirmed by the results in Figure 5, where the plating rate was evaluated after reducing $NiSO_4$ quantity by 15, 30, 45 and 60 wt%. from the standard MP formulation. Furthermore, EDS analysis on these coatings confirms that the decrease of $NiSO_4$ in solution is also associated with an increase of co-deposited P [16]. This result suggests that the availability of free nickel ions is a key factor in increasing the P content in the Ni-P alloy; however, it is also associated with a dramatic reduction in the plating rate.

Another strategy for obtaining a Ni-P coating with a higher P quantity consists in increasing the concentration of the complexing agent. Complexation is known to reduce the number of free nickel ions in the plating solution and change the mixed current potential, leading to both a reduction in the plating rate and changes in coating composition [39]. The effect of citric acid amount, with a fixed concentration of nickel sulfate and hypophosphite, was studied by increasing MP standard formulation by 10 wt%, 20 wt% and 30 wt%. The results in Figure 6 show that higher amounts of citric acid associate with a slower plating rate, only up to 20 wt% increase, where equi-molarity with $Ni^{2+}$ ions is achieved (i.e., all the citrate ions are bound to all $Ni^{2+}$ ions) [40]. Beyond this value, an increase in the plating rate is registered, and it can mainly be ascribed to the ΔpH induced by the presence of citric acid that is not bound to nickel ions [41]. A different trend is observed for P content, which is lower than the standard MP formulation for a citric acid increase of 10 wt% and 20 wt%. This is due to the increased chelation of $Ni^{2+}$ ions, which stabilizes the oxidized form of the metal and hinders its reduction, limiting the whole redox process [42]. When +30 wt% of citric acid is added, the lower pH promotes the deposition process and P deposition [1]. The addition of +40% of citric acid was tested, but the excessive increase in the complexing

agent concentration results in a large amount of adsorption onto the catalytic surface, with a poisoning effect that totally inhibits the deposition process [41,43].

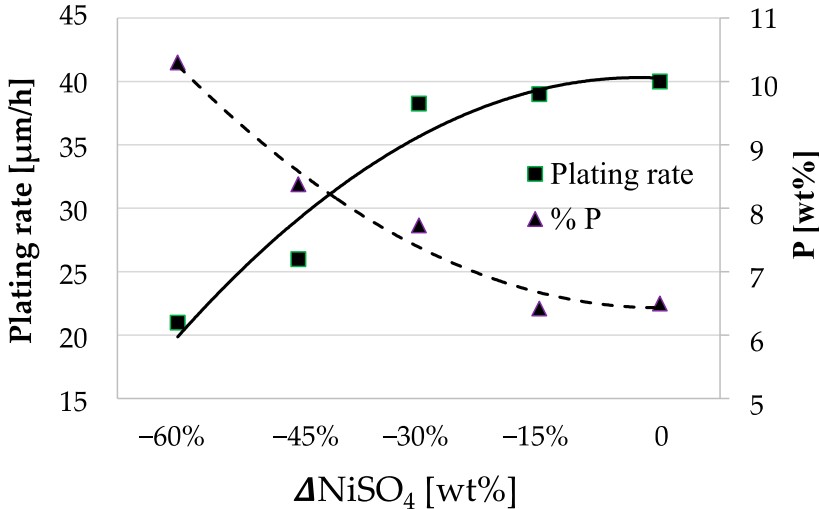

**Figure 5.** Variation of plating rate and P wt% as a function of $\Delta NiSO_4$ from MP formulation. Operating temperature: 90 °C. Dashed and solid lines are inserted for the readability of the graph and do not represent any mathematical relation.

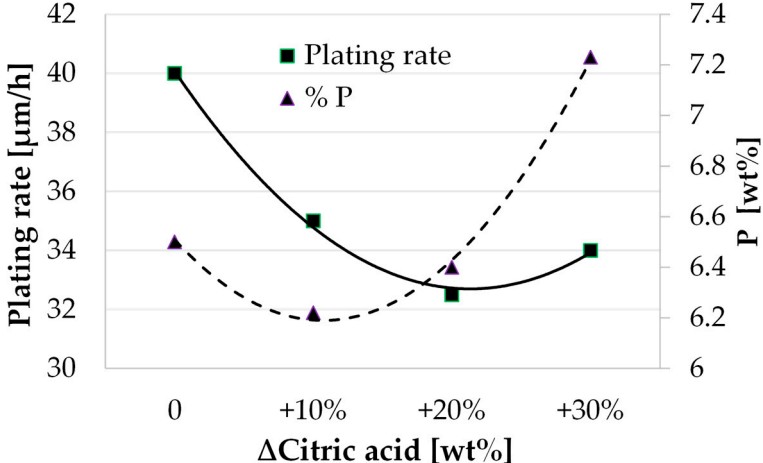

**Figure 6.** Plating rate and P wt% as a function of $\Delta$Citric acid from standard MP plating solution. Operating temperature = 90° C. Dashed, and solid lines are inserted for the readability of the graph and do not represent any mathematical relation.

Despite the promising result on P content increasing upon +30 wt% of citric acid, ~7 wt% of P is not sufficient to classify the coating as HP. For this reason, simultaneous variations of $NiSO_4$ and citric acid were studied to consider the cross-correlation between the two parameters. Variations from the MP standard formulation will be addressed with the nomenclature $MP_{x|y}$, where $\pm x$ represents the variation of $NiSO_4$ and $\pm y$ is the variation of citric acid. Figure 7 shows the results in a 3D histogram chart (mean results of 5 depositions of each experimental point are reported, the calculated standard deviation is always lower than 3% and is not reported in the figure for readability). The empty spots represent a combination for which the deposition cannot be activated; this can be traced back to the combined effect of decreasing availability of $Ni^{2+}$ ions when $NiSO_4$ concentration is decreased and the increased complexation upon the increasing amount of citric acid. The formulation with +10 wt% of citric acid and −45 wt% of $NiSO_4$ provides the best results in terms of plating rate (~30 µm/h) and %P (10.5 wt%). This confirms that the influence of reagents cannot be treated separately and that properties of the system

strongly depend on the mutual influence of every component: the amount of NiSO$_4$ has a strong influence on the amount of co-deposited P, but a good compromise with plating rate and bath stability can only be achieved by the concurrent optimization of the quantity of the complexing agent.

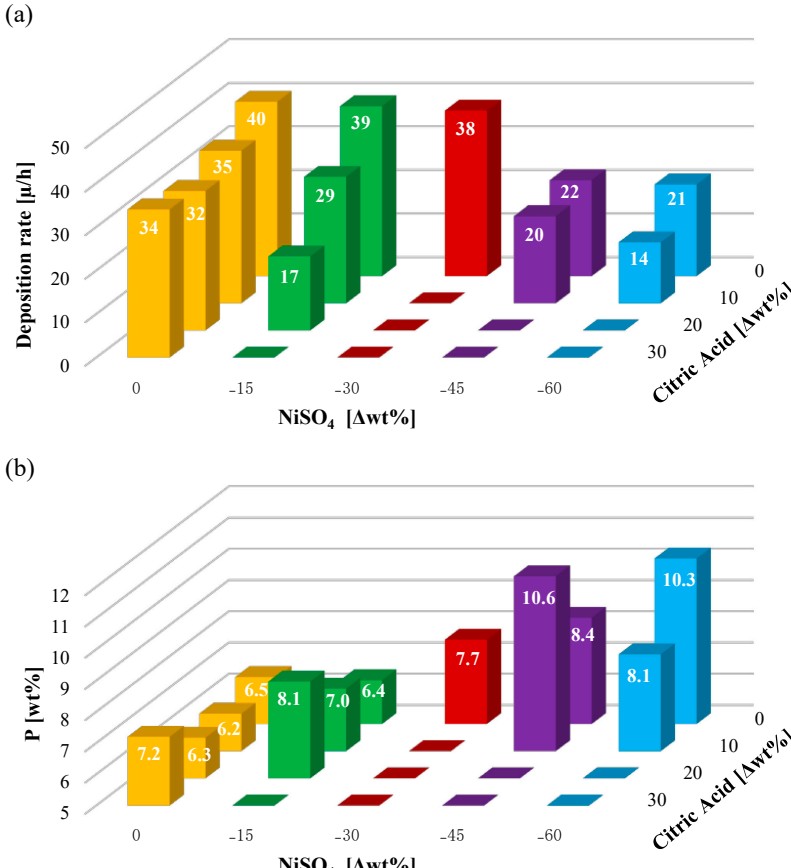

**Figure 7.** Plating rate and %P as a function of ΔNiSO$_4$ and a Δ Citric acid with respect to MP solution formulation. Operating temperature = 90 °C.

It should be remarked that the presented step of the experimental activity was conducted by keeping the concentration of the reducing agent constant, giving a variation in the [Ni$^{2+}$]/[H$_2$PO$_2^-$] addressed to the change in NiSO$_4$ concentration. To uncover the correlation between P wt% and the concentration of hypophosphite ions, the molar ratio [Ni$^{2+}$]/[H$_2$PO$_2^-$] was varied by changing the amount of the reducing agent, while all other reagents were kept constant. Concentrations of the hypophosphite ion were modified in order to vary the molar ratio [Ni$^{2+}$]/[H$_2$PO$_2^-$] in the range 1/25–1/3, and experimental activity was carried out on three formulations of the plating solution: (i) the standard MP formulation (MP$_{0|0}$), (ii) MP$_{-30|0}$, which shows intermediate properties of %P and plating rate, and (iii) MP$_{-45|+10}$, which guarantees the higher amount of co-deposited P.

Curves of %P variation are reported in Figure 8, and it can be noted that changing hypophosphite concentration leads to a non-linear variation of the amount of co-deposited P, with a similar trend for all the investigated formulations. This phenomenon can be traced by considering the reactions involved in the Ni-P deposition system [16]:

$$H_2PO_2^- + H_2O \ \rightarrow \ H_2PO_3^- + 2H^+ + 2e^- \tag{7}$$

$$Ni^{2+} + 2e^- \ \rightarrow \ Ni \tag{8}$$

$$H_2PO_2^- + 2H^+ + e^- \ \rightarrow \ P + 2H_2O \tag{9}$$

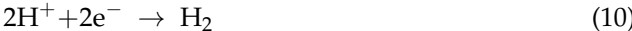

$$2H^+ + 2e^- \rightarrow H_2 \tag{10}$$

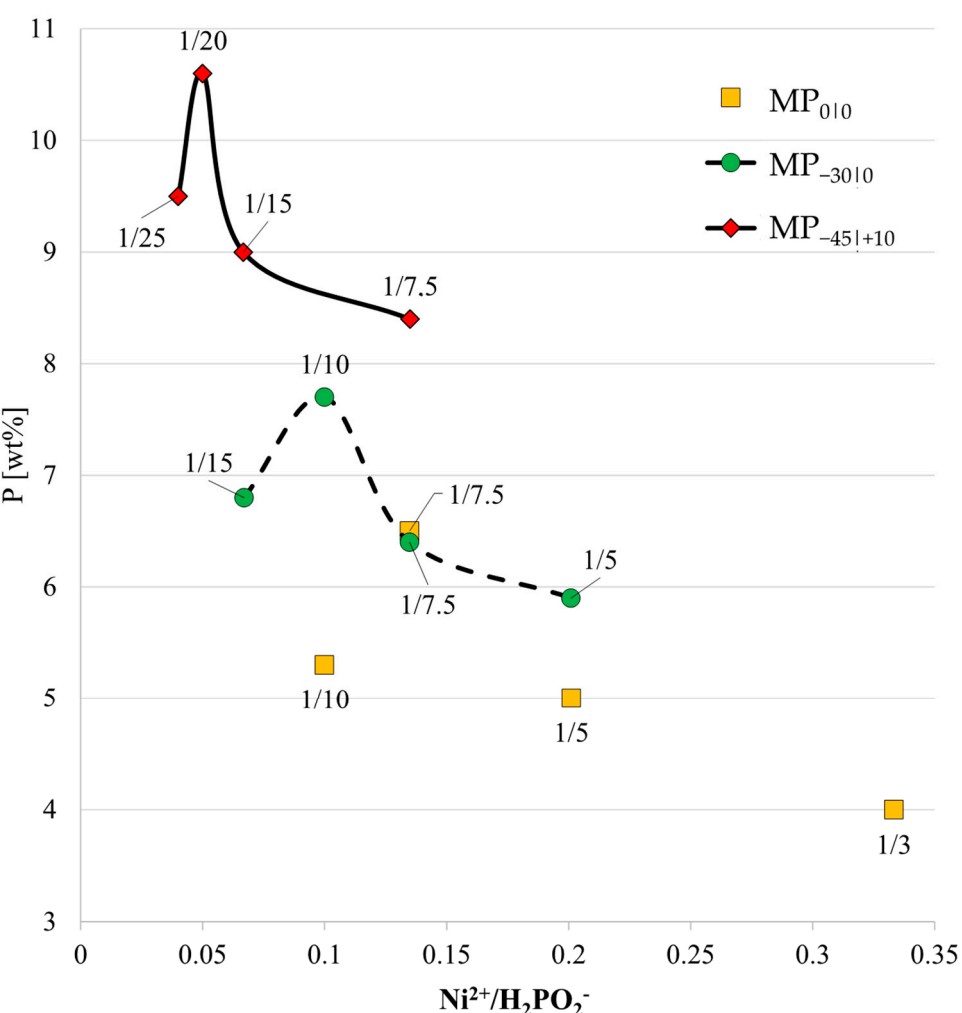

**Figure 8.** Phosphorous percentage (%P) as a function of $[Ni^{2+}]/[H_2PO_2{}^-]$. Operating temperature = 90 °C.

Reactions in Equations (7) and (8) refer, respectively, to the oxidation of hypophosphite and to the reduction of nickel; the reaction in Equation (9) describes the mechanism of phosphorus formation and its co-deposition into the alloy, whereas Equation (10) is a side reaction that leads to hydrogen evolution and makes no contribution to the deposition. The formation of the Ni-P matrix, given by Ni deposition and P co-deposition, can be considered a multi-step mechanism of parallel reactions that are interdependent on one another since reactions (7) and (8) are combined and (7) and (9) are combined. Even though a full understanding of the kinetics of the system goes beyond the scope of this work, Figure 8 depicts that P wt% in the coatings strongly depends on $Ni^{2+}/H_2PO_2{}^-$ ratio. Each curve was obtained with fixed $Ni^{2+}$ concentration and represented %P co-deposition according to $H_2PO_2{}^-$ variation. The amount of phosphorus in the coating with sodium hypophosphite could be illustrated according to the empirical kinetics equation [16,44]:

$$\frac{d[\mathrm{P}]}{dt} = \mathrm{k}\left[\mathrm{H_2PO_2^-}\right]^{1.91}\left[\mathrm{H^+}\right]^{0.25} \tag{11}$$

The amount of P in the coating increases with increasing concentration of hypophosphite ions, as could be expected and consistently with Equation (11), but only up to a threshold where the maximum of the curves is reached. A further increase in the quan-

tity of $H_2PO_2{}^-$ in the plating bath leads to lower amounts of co-deposited P. A similar behavior was observed by Sun et al. [44], who attributed this effect to the intensification of hypophosphite diffusion and adsorption, together with the increase in the proportion of hydrogen evolution. Each curve in Figure 8 follows a similar trend, with a maximum that is reached for progressively higher amounts of hypophosphite ions for formulation with a lower quantity of $Ni^{2+}$. Moreover, the maximum of %P is lower for higher concentrations of nickel in the plating solution, in accordance with the findings about the [$NiSO_4$] effect presented above. This last result confirms that the influence of each reagent cannot be evaluated separately from the rest of the system. The presented curves can represent a useful tool for the final optimization of the P wt% in the coating, according to the required properties given by the application.

It is worth mentioning that changes in the reducing agent concentration do not dramatically influence the deposition rate, as shown in Figure 9, and the main contribution is given by $Ni^{2+}$ availability.

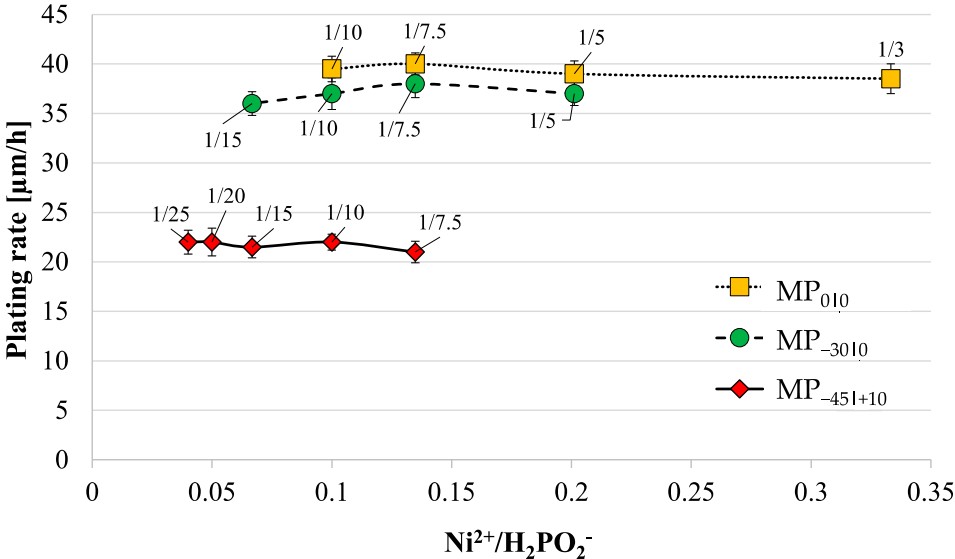

**Figure 9.** Deposition rate as a function of $[Ni^{2+}]/[H_2PO_2{}^-]$. Operating temperature = 90 °C.

The formulation $MP_{-45|+10|}$ with a molar ratio $[Ni^{2+}]/[H_2PO_2{}^-]$ equals 1/20 was selected as the best candidate for the HP formulation. However, it is poorly stable with respect to the standard MP solution, and adequate service life of the plating bath cannot be guaranteed. Therefore, the influence of increasing the concentration of the stabilizer (i.e., the thio-organic compound, or TOC) was studied to optimize its concentration in the HP solution. Results showing the plating rate and the wt% of P as a function of TOC concentration are reported in Figure 10. As similarly reported in the literature [16], the addition of the stabilizer does not dramatically affect the deposition rate, although a beneficial effect can be obtained when the concentration rises up to 8.5 ppm. Bath stability is guaranteed for concentrations in the range of 5–9 ppm: below 5 ppm, the solution is characterized by low stability and degrades at a temperature close to 90 °C, foreclosing the attainment of the best temperature for deposition; conversely concentrations higher than 9 ppm completely hamper the deposition to start. The last phenomenon is well-studied in the literature [30,45] and known as the "poisoning mechanism": sulfur from the thio-group of the TOC adsorbs on catalytically active sites to control deposition reactions, but when in excessive quantity, it can completely inhibit the process. Below the threshold of 5 ppm, the concentration of the stabilizer is too low to efficiently control the deposition; as the concentration increases, progressively fewer catalytic sites will be active for deposition, inhibiting the reduction of metal ions on undesired surfaces with high surface energy, like particles in solution or catalytic sites in the reactor/container, thus preventing random bath decomposition. A more efficient deposition on the substrate is therefore achieved and

associated with the increase in plating rate [24]. Nevertheless, when the concentration limit is exceeded, the poisoning mechanism takes place on the substrate itself so that it is no longer available as a catalytic surface, and the reaction does not start. The concentration of the stabilizer is also known to influence the P content in the deposit, leading to its increase when the cathodic reduction of nickel is inhibited [20,24,31]. It is important to point out that deposition with a TOC concentration of 5 ppm is characterized by a high plating rate and high P wt% but low stability that might lead to random bath decomposition. Therefore, 8.5 ppm was selected as the best quantity for the formulation of HP solution in order to ensure great stability of the plating bath, high plating rate and high wt% of P.

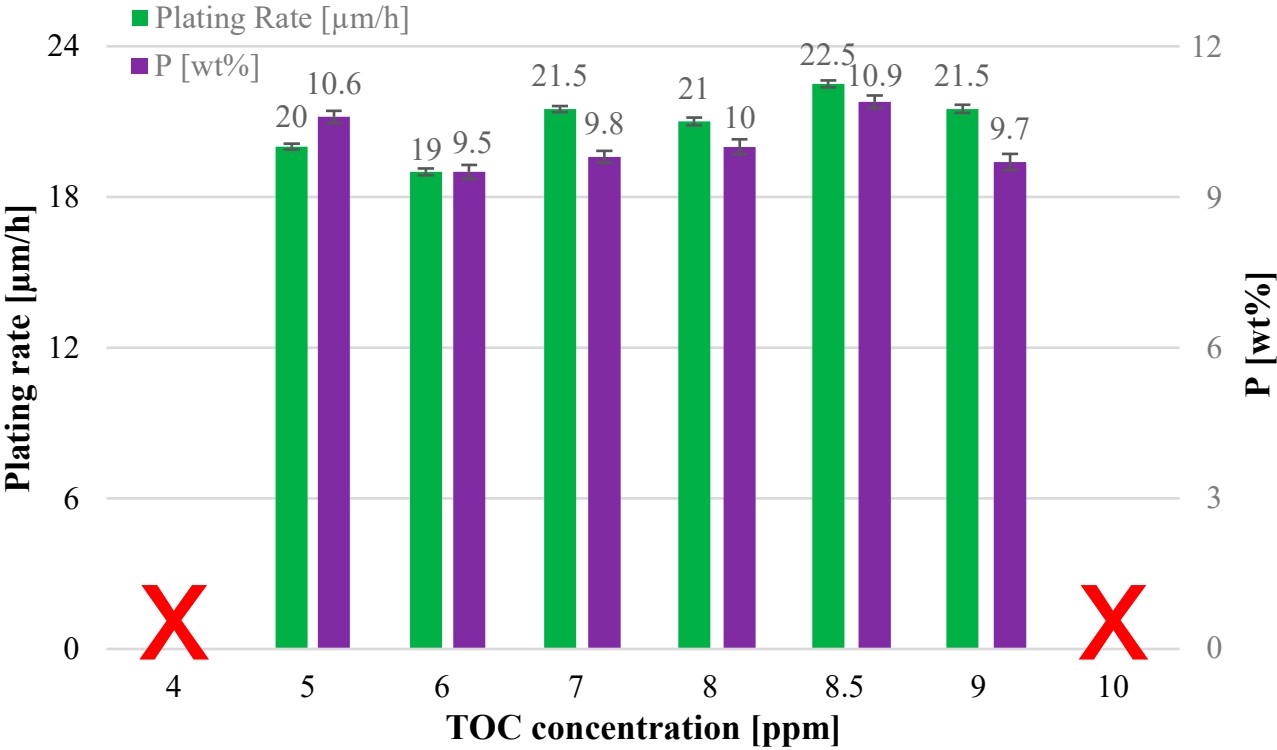

**Figure 10.** Plating rate and %P as a function of ppm of thio-organic compound. Plating temperature = 90 °C. $MP_{45.10-1/20}$ formulation.

To conclude, the study of all deposition parameters eventually led to the formulation of a stable solution for the deposition of HP coatings (10.9 wt% of P) and the composition is reported in Table 5.

**Table 5.** High phosphorous (HP) bath formulation obtained after optimization of the process parameters.

| Function | Name | Chemical Formula | HP (g/L) |
|---|---|---|---|
| Reducing agent | Sodium hypophosphite | $NaH_2PO_2$ | 51.5 |
| Buffer | Sodium acetate | $C_2H_3NaO_2$ | 15.0 |
| Chelating Agent | Citric acid | $C_6H_8O_7$ | 7.70 |
| Source of Nickel | Nickel sulfate | $NiSO_4$ | 6.60 |
| Stabilizer * | Thio-organic compound (TOC) | R-CS | 8.50 (ppm) |

* The stabilizer was added by liquid solution (1 mol/kg) in accordance with the quantity in ppm.

### 3.3. Coating Microstructure and Composition

The SEM surface micrographs of the MP and HP coatings are reported in Figure 11a,b, corresponding to the XRD microstructural analysis and EDS elemental analysis presented

in Figure 11c and 11d, respectively. Both samples exhibit a nodular morphology with a cauliflower-like structure, typical of coatings obtained by the electroless plating technique [46,47]; however, HP coatings show a denser network with a smaller size of individual nodules than the MP. This effect can be addressed by the higher quantity of stabilizer in the HP solution, which limits lateral growth [24,31,48]. The comparison of the two XRD patterns shows that MP coatings are characterized by a higher degree of crystallinity compared to HP ones. These characteristics are expected because the higher P content in the Ni matrix distorts the lattice to an extent where amorphous nickel is obtained [49], whereas P content in MP coatings is only sufficient to refine grain size and create a nanocrystalline structure [14]. Yet, the amorphous nature of HP coatings is responsible for their high corrosion resistance [50,51]. All the peaks in the XRD spectra (45°, 52° and 75°) refer to cubic Ni (JCPDS 65-2865) (Figure 11). The use of thio-stabilizers is sometimes associated with S contamination in the coatings, with negative effects on corrosion resistance [52]. Nevertheless, the EDS spectrum reveals the presence of Ni and P only, confirming the purity of the obtained coatings. The quantitative EDS analysis performed on the top view of the two coatings shows content of P equal to $10.9 \pm 0.3$ wt% for the HP and $6.5 \pm 0.2$ wt% for the MP.

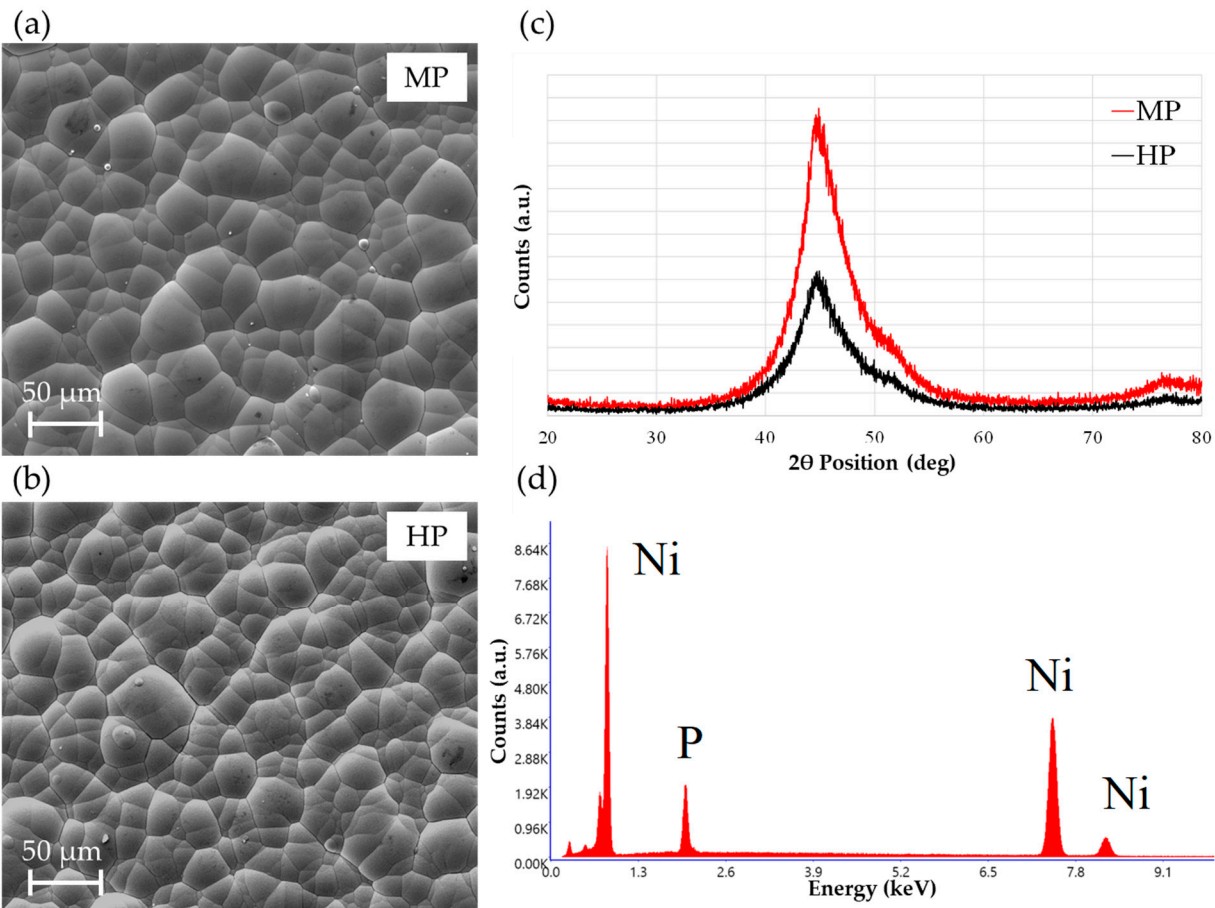

**Figure 11.** Morphological and elementary analysis for MP (6.5 wt%) and HP (10.9 wt%) coatings obtained by top-view SEM images (**a**,**b** respectively), X-ray diffraction (**c**) and EDS (**d**).

### 3.4. Pull Off-Test

Pull-off tests were performed to compare the adhesion strength of MP and HP coatings obtained after simple pre-cleaning (soaking in NaOH and sandblasting) and after the standard pre-treatment process (pre-cleaning and activation by pickling with HCl 50%, as defined in Section 3.1). During an adhesion test, detachment between the coated rod and the uncoated one (counterpart) can happen by different mechanisms: (i) detachment

between the glue and the counterpart; (ii) detachment between the glue and the coated road; (iii) detachment between the coating and the substrate; (iv) cohesive failure of the coating. These different kinds of rupture are not necessarily independent and might also occur simultaneously [53]. The tested samples coated after the standard pre-treatment always experienced a cohesive/adhesive separation between the glue and the counterpart, as shown in Figure 12a, whereas samples coated just after pre-cleaning typically showed rupture by a partial detachment of the coating (Figure 12b).

(a)

(b)

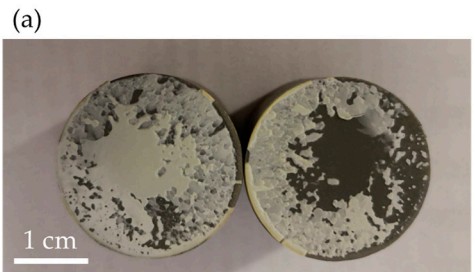
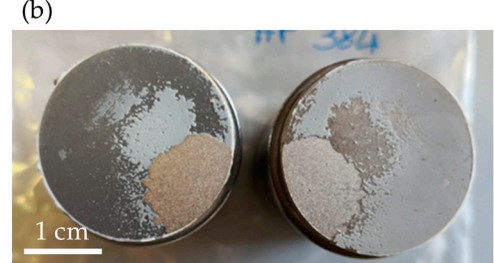

**Figure 12.** Example of adhesive/cohesive glue detachment in pre-treated samples (**a**) and partial detachment of the coating when no pre-treatment was performed (**b**).

Accounting that the reference value for the bonding strength of the adhesive film was $80 \pm 1$ MPa, the values of the maximum load experienced by the samples before rupture are reassumed in Figure 13. Both MP and HP coatings obtained after standard activation pre-treatment are characterized by better adhesion compared to those that underwent pre-cleaning only. Moreover, the samples coated without the activation procedure are characterized by lower adhesion and exhibit failure by a partial detachment of the coating from the substrate, probably due to the incomplete removal of oxides or products from long-term storage of substrates. Surface activation is known to have an important role in adhesion strength between the coating and the substrate [54,55] since they affect the mechanism of nucleation and growth of the Ni-P deposit. The presence of impurities, such as greases, sandblasting residues and /or oxide layers, may favor the formation of pores and cracks at the coating-substrate interface, thus decreasing overall adhesion. Therefore, both results about the maximum load before rupture and analysis of the failure mechanism enlighten the better adhesion of the coating when a standard activation pre-treatment is carried out.

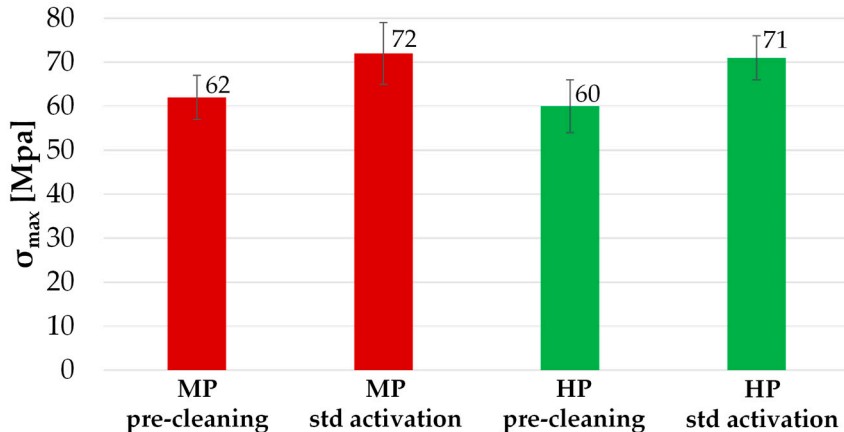

**Figure 13.** Pull-off test for the MP and HP samples obtained with standard activation treatments compared with those obtained with pre-cleaning only. The reference value of the adhesive film is 80 MPa.

### 3.5. Heat Treatments

Heat treatments above the crystallization temperature of the Ni-P alloy (reported to be around 340 °C [34,56,57]) are often employed to modify coating properties; therefore, the evolution of coating characteristics upon high-temperature exposure at 400 °C and 600 °C for 1 h in the air was investigated. All the samples involved in the characterization process are summarized in Table 6.

**Table 6.** Samples (MP and HP) for morphological, microstructural and mechanical characterization.

| Sample | wt% P | Heat Treatment | Activation |
|---|---|---|---|
| $MP_{AC}$ | 6.5 | - | HCl 50 vol.%, 1 min |
| $MP_{400}$ | 6.5 | 400 °C, 1 h | HCl 50 vol.%, 1 min |
| $MP_{600}$ | 6.5 | 600 °C, 1 h | HCl 50 vol.%, 1 min |
| $HP_{AC}$ | 10.9 | - | HCl 50 vol.%, 1 min |
| $HP_{400}$ | 10.9 | 400 °C, 1 h | HCl 50 vol.%, 1 min |
| $HP_{600}$ | 10.9 | 600 °C, 1 h | HCl 50 vol.%, 1 min |

SEM micrographs in Figure 14 show the morphological evolution of samples according to heat treatment temperature. Surface morphology of MP coatings in the as-deposited condition, after annealing for 1 h at 400 °C and 600 °C are reported in Figure 14a, 14b and 14c, respectively; the same surface condition for HP coatings are reported in Figure 14d–f, respectively. It can be observed that the thermal treatment at 400 °C does not induce particular morphological modifications; conversely, exposure at 600 °C leads to changes in morphology and an oxide scale visibly begins to grow on the surface.

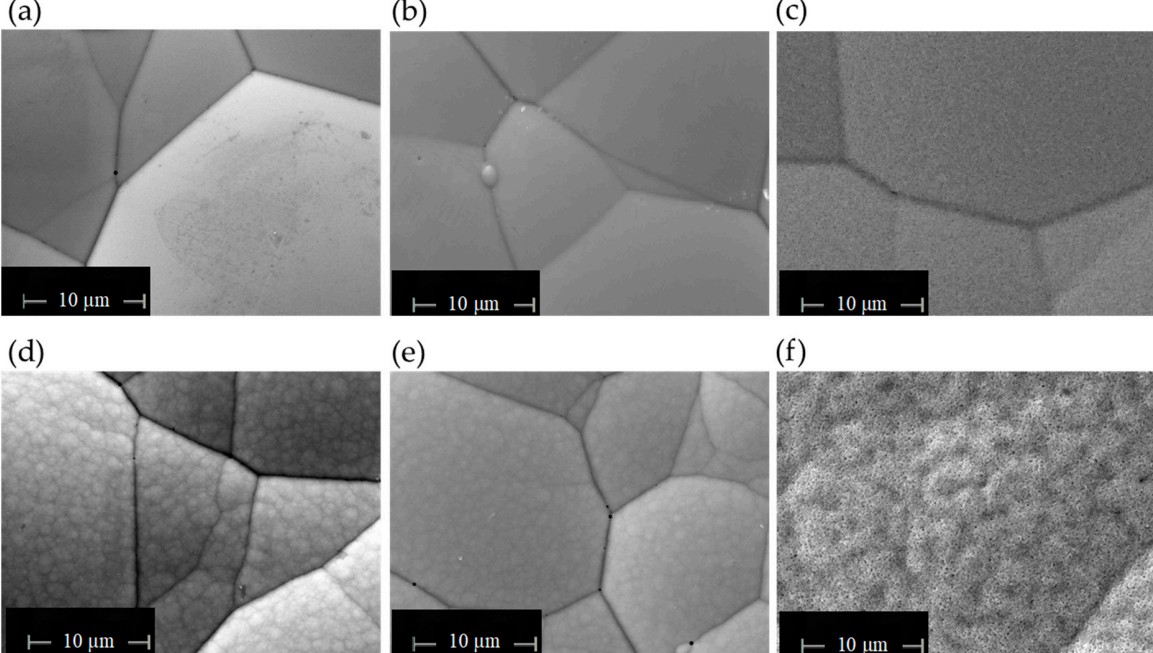

**Figure 14.** Morphological analysis (SEM top-view) for (**a**) MP as deposited; (**b**) MP after annealing at 400 °C for 1 h in air; (**c**) MP after annealing at 600 °C for 1 h in air; (**d**) HP as deposited; (**e**) HP after annealing at 400 °C for 1 h in air; (**f**) HP after annealing at 600 °C for 1 h in air.

XRD spectra in Figure 15 show the microstructural evolution of MP (a) and HP (b) coatings as a function of heat treatment temperature. Substantial changes in microstructure are observed in both MP and HP coatings after treatment at 400 °C: recrystallization of the f.c.c. Ni matrix is observed together with precipitation of b.c.t. Ni₃P [49,58–62]. A segregation process of phosphorus at the grain boundaries and triple junctions of the Ni-

P grains occurs with increasing temperature, leading to the formation of P-rich regions [46] and precipitation of $Ni_3P$ when P concentration exceeds a certain threshold (reported to be around 15 wt% of P) [63]. There is evidence that the $Ni_3P$ precipitation preferentially occurs at grain boundaries and triple junctions [64,65]; therefore, MP coatings (nanocrystalline in the as-deposited condition) experience a continuous phosphorus segregation process when temperature increases. However, Farber et al. [46] proposed that structural changes upon heat treatments depend on the formation of a metastable grain boundary phase that already develops in the as-deposited state of HP coatings. For this reason, precipitation of the $Ni_3P$ compound is thought to have taken place earlier in the HP deposits [34,66], and a massive presence of precipitates can be observed on the HP spectrum, even though they are characterized by lower crystallinity than MP coatings. In fact, sharper Ni peaks can be observed for MP, which associates with a greater grain size of the matrix. After treatment at 600 °C, both spectra become sharper, indicating grain growth of both Ni and $Ni_3P$ phases. NiO peaks also appear, confirming the presence of a thin scale of oxide, as observed by SEM images. The finer morphology of HP coatings and their lower surface roughness are expected to influence the growth of the thin oxide film and can also be responsible for the different morphology of the scale. The evolution of average grain size, calculated with the Scherrer equation (Equation (1)), and coarsening of $Ni_3P$ precipitates with increasing temperature is reported in Figure 16.

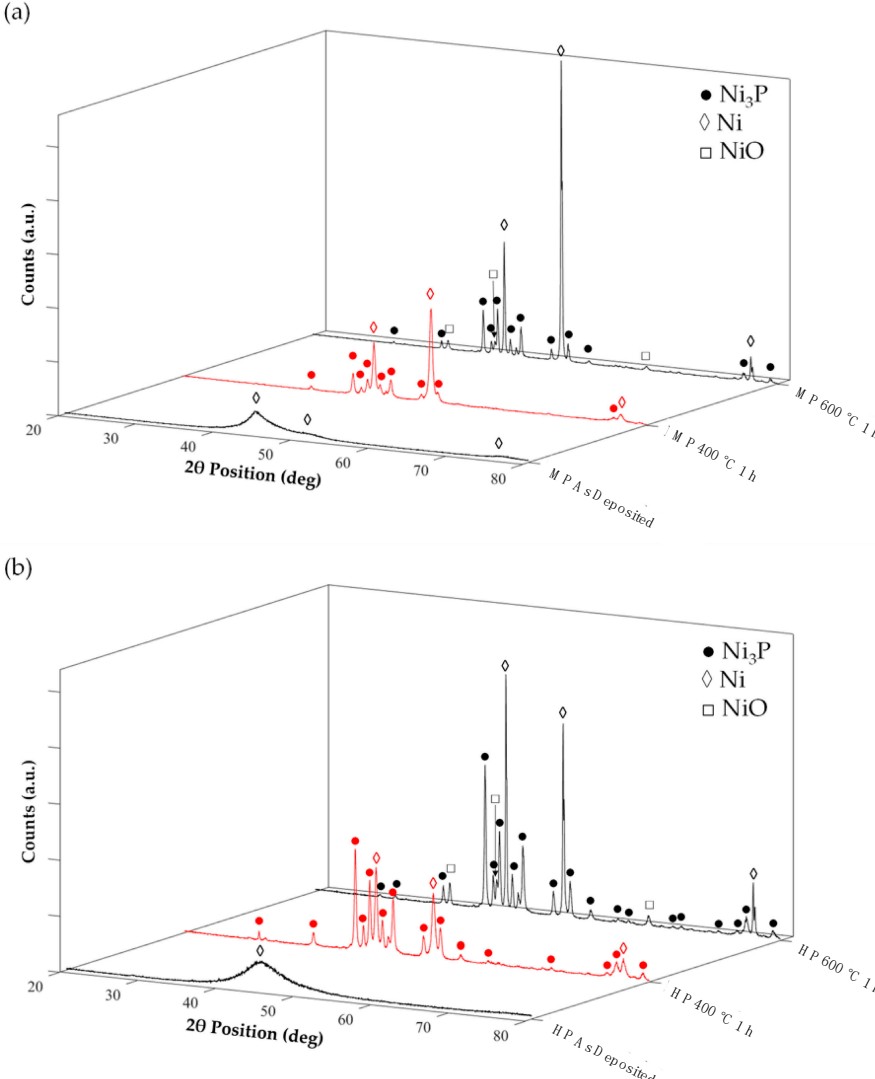

**Figure 15.** XRD spectra for MP (**a**) and HP (**b**) coatings showing microstructural evolution upon heat treatments for 1 h in air.

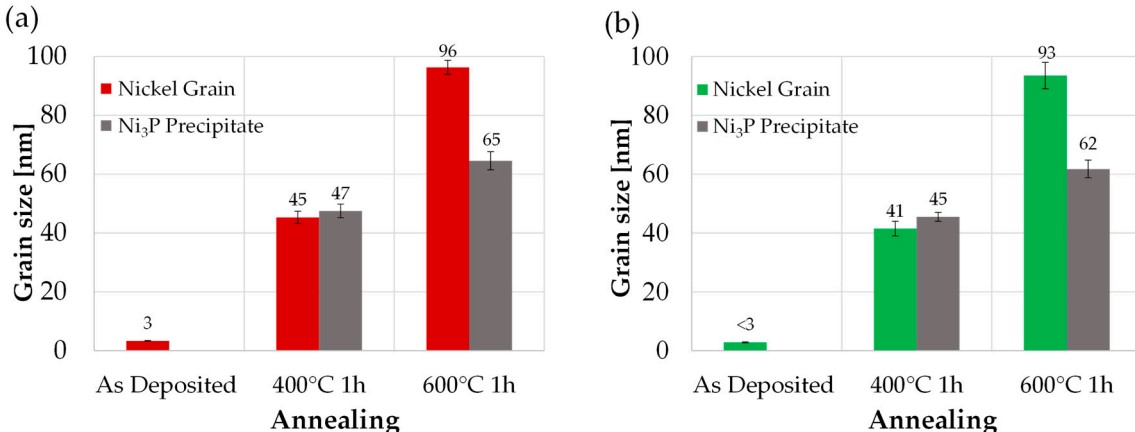

**Figure 16.** Increase of average nickel crystallite size and coarsening of $Ni_3P$ precipitates with increasing annealing temperature for MP (**a**) and HP (**b**) coatings.

No trace of superficial or internal cracks induced by the heat treatment was observed by SEM analysis, and samples were tested negative for the ferroxyl reagent test.

### 3.6. Vickers Hardness

The microhardness of MP and HP coatings in the as-coated condition and after heat treatments are shown in Figure 17. The nanocrystalline structure of deposited MP coatings guarantees higher hardness than amorphous HP. As shown in Figure 16, heat treatment at 400 °C is associated with increased crystallite size of nickel and precipitation of hard $Ni_3P$ phases, which lead to hardness increase of the deposits. In contrast, a dramatic hardness decrease is registered after treatment at 600 °C due to excessive grain growth and the formation of coarse precipitates. Deformation processes of deposited Ni-P coatings, which have very fine grain size or can even be amorphous, show a reverse Hall-Patch effect [67]; therefore, grain growth induced by heat treatment at 400 °C associates with lower grain boundary sliding and rotation [68,69] and results in increased microhardness. The presented results are in accordance with literature data [49,60,61,70,71] that report maximum hardness after heat treatments at 400 °C where $Ni_3P$ precipitates guarantee a precipitation hardening effect and grain growth is limited to an extent that causes strengthening. Conversely, the samples treated at 600 °C exhibit lower microhardness values because of the excessive grain growth and the formation of coarse precipitates with a less effective hardening effect in accordance with the Orowan strengthening mechanism [72].

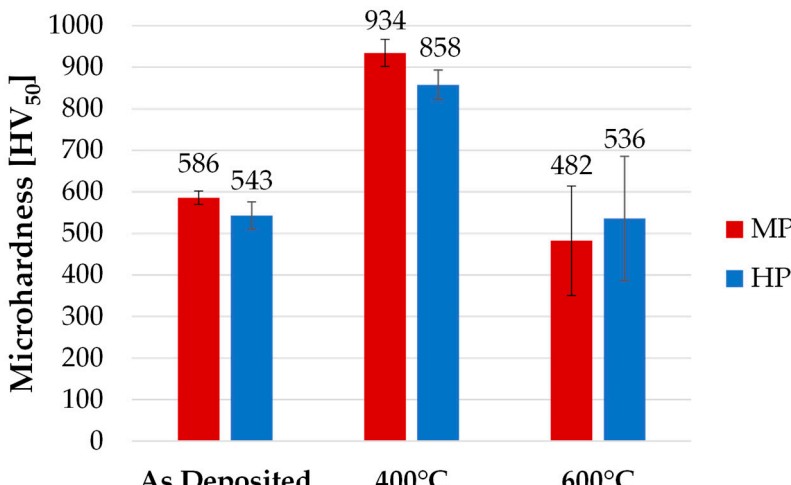

**Figure 17.** Microhardness (HV) for the MP and HP coatings in the as-deposited condition and after heat treatment at 400 °C and 600 °C for 1 h in air.

## 4. Conclusions

A new lead-free electroless nickel plating process was developed for the plating of MP coatings and optimized to obtain a formulation for HP deposition. Pre-deposition treatments are found to be crucial for the activation of deposition and in determining the adhesion strength of the coatings, with the best results of plating rate obtained when performing a pre-treatment of pickling with HCl 50%vol for 1 min. A study on the chemical composition of the solution demonstrated that the ratio between the reducing agent and metal ion is a key parameter for the tailoring of the P co-deposition. In particular, decreasing the $Ni^{2+}/H_2PO_2^-$ ratio by a reduction of 40 wt% of $NiSO_4$ and increasing of 10 wt% of complexing agent leads P co-deposition up to 10.6 wt%. Further optimization of the TOC allows the P content to rise up to 10.8 wt%, while guaranteeing the best results on bath stability. Best deposition parameters guaranteed a plating rate of 40 μm/h for the MP coatings and 25 μm/h for HP coatings, with deposition at 90 °C. Characterization of coatings confirmed the nanocrystalline and amorphous nature of MP and HP coatings, respectively, with hardness up to 930 HV and 840 HV when heat-treated at 400 °C for 1 h. No through-thickness cracks were detected by the ferroxyl reagent tests performed on the samples treated at 400 °C.

**Author Contributions:** Conceptualization, A.P., M.R. and F.M.; validation, G.P. (Giovanni Pulci); formal analysis, V.G. and L.P.; investigation, V.G., L.P. and G.P. (Giulia Pedrizzetti); data curation, G.P. (Giulia Pedrizzetti); writing—original draft, V.G.; writing—review and editing, G.P. (Giovanni Pulci) and G.P. (Giulia Pedrizzetti); supervision, G.P. (Giovanni Pulci) and F.M.; project administration, F.M.; funding acquisition, A.P. and M.R. All authors have read and agreed to the published version of the manuscript.

**Funding:** This research received no external funding.

**Institutional Review Board Statement:** Not applicable.

**Informed Consent Statement:** Not applicable.

**Data Availability Statement:** Not applicable.

**Conflicts of Interest:** The authors declare no conflict of interest.

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
