# Peer review of "Medium and High Phosphorous Ni-P Coatings Obtained via an Electroless Approach: Optimization of Solution Formulation and Characterization of Coatings"

_coatings, doi:10.3390/coatings13091490_

Round 1

Reviewer 1 Report

Surface treatment plays an important role in improving the properties of materials. In this work, an electroless nickel plating process was developed for the plating of MP coatings and optimized to obtain a formulation for HP deposition. Pre-deposition treatments are found to be crucial for the activation of deposition and in determining adhesion strength of the coatings. This study is very meaningful. The main comments are as follows:

1. Line 24, The number 3 in “Ni3P  should be the subscript.

2. Line 100, line 104, line 126, Ni2+/H2PO2- should added the subscript and superscript.

3. The superscript in Table 1 is not standard.

4. Such signs appear in many places: Error! Reference source not found. The authors should check the errors.

5. Figure 16 should add the error of grain statistics.

Author Response

Please, see attachment.

Reviewer 2 Report

Comments of coatings-2536921

There is a deep review of manuscripts related with the study. The discussion of the presented results is good. Since the manuscript seems to me to be quite clear and the discussion and presentation of the results is good, I have no important observations to make about the paper. Therefore, my suggestion is that it be published in its present form.

Author Response

The author thank the reviewer.

Reviewer 3 Report

In the present study, the authors developed a lead-free electroless NiP plating solution for the deposition of coatings with medium phosphorus content (6-9 wt%) and they optimized the composition to obtain deposits with high phosphorus (10-14 wt%). Also, they studied the cleaning and activation treatments in terms of effectiveness and influence on the deposition rate. Finally, the authors investigated the concentration of reagents (nickel salt, complexing agent, reducing agent and stabilizer) and their combined effect on P content and plating rate.

The issue is attractive due to it could be a promising technique for many technical applications.

The paper could be published after the authors attend the comments and suggestion.  

- Page 5. “Table 1” should be “Table 2” and the authors should be state all the chemical formulas in appropriate form.

- Page 6, lines 251-253. “The effect of surface pre-deposition treatments (referred as pre-treatments) was studied varying the conditions of acid pickling and results are reported Error! Reference source 2not found.”. It is not clear if the effect of the surface pre-deposition treatments studied varying the conditions of acid pickling was already reported or the results are reported in the present work.

- Page 6, lines 256. “…..last column of the table.”, which table?, it needs to be specify.

Figure 1. The legend in X-axis is wrong, please corrected in the appropriate form.

- Page 6, reactions (1) and (3) are not correctly balanced.

- Page 7, lines 293-295. The authors state “To better understand this phenomenon, substrates were oxidized at 400°C and 600°C for 4 hours, to simulate different conditions of degradation.”. The authors should be state how the samples were oxidized.

- Table 3 has different significant figures in which the number of digits in a value, please corrected.

- Page 9, “Figure 1” should be “Figure 2”, and the legend in X-axis is wrong, please corrected in the appropriate form.

- Page 11, reactions (5) is not correctly balanced.

- Page 17, Table 5. The authors should be state all the chemical formulas in appropriate form. Also, the 3 column has different significant figures in which the number of digits in a value, please corrected.

- Page 18, figure 11. Place separate Figure 11 into Figure 11.a, 11.b, 11.c and 11.d. Figure of X-ray diffraction has the lagend X-axis wrong and EDS figure does not have a legend on any axis.

- Page 23, lines 641-642. The phase “Study on the chemistry of the solution demonstrated that decreasing Ni2+/H2PO2 ratio by reduction of 40 wt% of NiSO4 and increasing of 10%wt of complexing agent leads P co-deposition up to 10.6 wt%.”. It not clear, what do the authors want to conclude?  

General comments:

- “Error! Reference source not found” appears throughout the manuscript, it is very very confusing due to sometimes it referred to references and in other time figures and so on, please add the reference appropriately.

- In some places and figures the authors state “10 minutes” in other “10´ in” please standardize in the standard units.  

- chemical formulas appears throughout the manuscript inappropriately, please correct them.

- All tables and figures should be state in the text of the manuscript.

- Check all references, they should include year, voume, pag., doi, etc.

Minor editing of English language required.

Author Response

Please, see attachment.

Reviewer 4 Report

Paper: Medium and high phosphorous Ni-P coatings obtained via electroless approach: optimization of solution formulation and characterization of coatings presents some interesting experimental results in the field of coatings. However the article presents some problems that can be addressed before publication. 

The sentence : Error! Reference source not found.. is all over the article, not sure if is the authors fault however it makes it difficult to read and analyze the article 

In Abstract section the sentence: The obtained coatings were analyzed by SEM and XRD and thermally treated at 400°C and 600°C to study microstructural and hardness evolution - must be rewritten and explain if the authors use SEM technique to evaluate the hardness indentation trace or how they evaluate the hardness with SEM and XRD 

L21: re-phrase : The best deposition parameters were defined, allowing deposition of MP coatings (6.5 wt% of P) with plating rate of 40 μm/h and HP coatings (10.9 wt% of P) with plating rate of 25 μm/h at 90°C - it s hard to understand the deposition process 

The references groups are not recommended at general presentations , use only one or two representatives studies - for example : at the phrase: The flexibility of the solution chemistry allows its tuning to investigate innovative and cheaper formulations, to meet specific needs of the final product - the authors can find at least 100 titles ... 

The authors should use subscript and superscript all over the paper , some times is confusing not to. 

The introduction section is too long 

The paper has two Table 1 , at line 210: should be  Table 2,  in text, Line 183 is ok 

Quality of Figure 3 must be improved 

A scale in Figure 12 can be helpful. 

In figure 14 explain each figure what it is 

The authors presents some interesting results in : Figure 16. Increase of average nickel crystallite size and coarsening of Ni3P precipitates with increasing annealing temperature for MP (a) and HP (b) coatings. - are these results extracted from figure 14 results : HP? or only XRD  results ? more comments and explanations are necesary in this case , also  the results are not clear : 3 or 47nm sharp ?, provide a St Dev value for each determination 

The conclusion section can be re-structured to highligth the main reults and findigs 

The references section is proper. 

Author Response

Please, see attachment.

Reviewer 5 Report

This is a high-quality engineering research paper, which reports a new lead-free electroless deposition technique of Ni-P coatings. In this work, the concentration of reagents and cleaning/activation treatments were controlled to study the influence on the deposition quality and rate, meanwhile, the effects of phosphorus content and heat treatment on the physicochemical properties of the coating were also studied. Because the content of this paper is systematic and detailed, the research results are reproducibility, deposition technology has a high potential for engineering application. Although there is a lack of some scientific discussion and some errors in the manuscript that need to be corrected, it basically meets the requirements for publication. Specific comments are included below.

Generally:

1. Some paragraph and figure format need to be revised and unified, you can refer to the format of other Coatings published papers.

2. Perhaps because the authors used plug-in software to edit the manuscript, many Figures information in the text cannot be displayed (shown as “Error! Reference source not found.”), please revise.

3. Lines 316-317: the Figure 1 sequence number is wrong, the description here should be Figure 2.

4. The ordinal number of the secondary heading is mislabeled, there are many 3.2.

5. Due to the large number of parallel samples involved in the study, it is appropriate to add a Table in Materials and Methods section with sample name, abbreviation, and treating technology, so that the readers can refer to it more clearly and quickly.

Specifically

6. In Introduction section Compared with the electrodeposition method, electroless plating guarantees several advantages that make it a promising technique for many technical applications: Please provide some specific application cases or specific potential applications.

7. Lines 188-192: Authors describe that pH strongly influences the P content and the plating rate and cite references. If the pH value is a necessary parameter, and no research is required, please add explanations where appropriate.

8. In this study, a multi-stage surface pretreatment process was used, one of which was aimed at producing surface activation. What is the influence mechanism of surface activation on coating preparation? Is the substrate oxide film the main factor affecting surface activation? Please discuss in section 3.1.

9. Lines 350-352: Authors describe that too slow deposition rates may limit applicability when thick coatings are required, please add an explanation of the relationship among deposition rate, coating thickness, and coating applicability.

10. Lines 571-574: 340℃ is the recrystallization temperature of what material? That is important to clarify.

11.Figure 14 and 16: In Figure 14, large grain sizes (> 10 μm) and even recrystallization can be observed, and the large grains contain subgrains. However, the calculated results in Figure 16 are different greatly from the observed results in Figure 14, please explain why and discuss.

  • The quality of the English language is no problem.

Author Response

Please, see attachment.

Round 2

Reviewer 3 Report

Authors have addressed all the concerns/suggestions from the previous review. The paper can be published in Coating.

Reviewer 4 Report

The authors respond to the revisions, the images quality was improved, paper can be publish in the current form.